# On the Optimal Efficiency of A$^*$ with Dominance Pruning

## Álvaro Torralba

Department of Computer Science, Aalborg University, Denmark
alto@cs.aau.dk

## Abstract

A well known result is that, given a consistent heuristic and no other source of information, A$^*$ does expand a minimal number of nodes up to tie-breaking. We extend this analysis for A$^*$ with dominance pruning, which exploits a dominance relation to eliminate some nodes during the search. We show that the expansion order of A$^*$ is not necessarily optimally efficient when considering dominance pruning with arbitrary dominance relations, but it remains optimally efficient under certain restrictions for the heuristic and dominance relation.

## Introduction

Heuristic best-first search algorithms are a fundamental tool for problem solving whenever the problem can be modeled as finding paths in graphs. Heuristic functions guide the search towards the goal by estimating the distance from any given state to the goal. Whenever an optimal solution of minimum cost is required, A$^*$ search is often the algorithm of choice (Hart, Nilsson, and Raphael 1968). This is well supported by the well known result that, given a consistent heuristic $h$ and no other source of information, A$^*$ does expand a minimal number of nodes up to tie-breaking among all algorithms that guarantee finding the optimal solution (Dechter and Pearl 1985).

Dominance pruning is a technique to eliminate nodes during the search if they can be proven to be dominated by another state (Hall et al. 2013; Torralba and Hoffmann 2015). This exploits an additional source of information in the form of a dominance relation $\preceq$, which compares two states to determine whether one can be proven to be as close to the goal as the other. This type of dominance appears naturally on problems that have to deal with resources, (i.e., removing states that have strictly less resources than another), and can also be applied on other kinds of problems (e.g., in gridworlds being at a central square can sometimes be proven better than being at a corner if the set of reachable squares in one step is strictly larger). This can be exploited by any search algorithm to reduce the number of nodes explored while retaining any solution optimality guarantees. This has been mainly used in the context of cost-optimal planning, as an enhancement for the A$^*$ algorithm.

In this paper, we address the question of whether the expansion order of A$^*$ is good to minimize the number of expansions when dominance pruning is used. Prioritizing the expansion of states with lower $f$-value is not necessarily an obvious choice anymore, since states that are more promising according to the heuristic function are not necessarily better according to the dominance relation. Furthermore, previous results proving the optimal efficiency of A$^*$ are no longer valid due to having a new source of information.

Indeed, we show that there are cases where A$^*$ with dominance pruning is not optimally efficient, and that different expansion orderings, or even expanding some states that could be pruned lead to a globally higher number of expansions in some cases. However, these cases can be attributed to "inconsistencies" in the information provided by the heuristic function and the dominance relation. We extend the notion of consistent heuristics to consistent heuristic and dominance relation pairs, and prove that A$^*$ with dominance pruning is indeed optimally efficient, meaning that there is a tie-breaking for A$^*$ that is guaranteed to expand the lowest number of nodes among all algorithms with admissible heuristics and dominance pruning.

We also analyze which tie-breaking strategies remain optimally efficient up to the last $f$-layer, i.e., when we ignore the expansions of nodes with an $f$-value equal to the solution cost. This is relevant because when consistent heuristics are used, the choice of tie-breaking rule in A$^*$ is only relevant for the last layer, since all nodes with an $f$-value lower than the optimal solution cost must be expanded regardless of the expansion order. Therefore, most implementations of A$^*$ choose tie-breaking strategies in favor of nodes with lower $h$-value, which are expected to find a solution faster in the last $f$-layer. We show that with dominance pruning this is no longer the case, as tie-breaking strategies in favor of nodes with lower $g$-value are preferable up to the last layer.

## Background

A *transition system* (TS) is a tuple $\Theta = \langle S, L, T, s^I, S^G \rangle$ where $S$ is a finite set of *states*, $L$ is a finite set of *labels* each associated with a *label cost* $c(l) \in \mathbb{R}_0^+$, $T \subseteq S \times L \times S$ is a set of *transitions*, $s^I \in S$ is the *start state*, and $S^G \subseteq S$ is the set of *goal states*. We write $s \xrightarrow{l} t$ as a shorthand for $(s, l, t) \in T$. A *plan* for a state $s$ is a path from $s$ to any $s_G \in S^G$. We use $h^*(s)$ ($g^*(s)$) to denote the cost of a cheapest plan for $s$ (path from $s^I$ to $s$). A plan for $s$ is *optimal* iff its cost equals $h^*(s)$. The sum $f^*(s) = g^*(s) + h^*(s)$ is the cost

of an optimal plan from $s^I$ passing through $s$. We denote $F^*$ to the optimal solution cost for $s^I$, $F^* = f^*(s^I) = h^*(s^I)$.

To deal with tasks with 0-cost actions, we define a modified cost function $c_\epsilon$ so that all 0-cost actions are assigned a cost of $\epsilon$, where $\epsilon$ is a tiny constant such that the sum of arbitrarily many $\epsilon$ will still be lower than any other action cost greater than 0. We define $g_\epsilon, h_\epsilon^*, f_\epsilon, \ldots$ as the functions above under this new cost function.

A *heuristic* $h : S \mapsto \mathbb{R}_0^+ \cup \{\infty\}$ is a function that estimates goal distance. It is *admissible* if it never overestimates the real cost, i.e., $h(s) \le h^*(s)$ for all $s \in S$, and *consistent* if for any $s \xrightarrow{l} t$ it holds that $h(s) \le h(t) + c(l)$.

Best-first search algorithms maintain an open and a closed list with all nodes that have been seen so far. A search node $n_s$ characterizes a path from the initial state to the final state of the path, $s$, where the $g$-value $g(n_s)$ is the cost of the path. We write $n_s \xrightarrow{l} n_t$ as a shorthand for $s \xrightarrow{l} t$ and $g(n_t) = g(n_s) + c(l)$. The open list is initialized with the initial state that has a $g$-value of 0. At each step, a node is selected from the open list for expansion. When a node is expanded, it is removed from open and all the successors are generated and inserted into the open list. The closed list keeps all nodes that have been expanded to avoid duplicates so that a node is not expanded if another node with the same state and a lower or equal $g$-value has already been expanded. A$^*$ always selects for expansion a node with minimum $f$-value where $f(n_s) = g(n_s) + h(s)$. Since the behavior of A$^*$ is not uniquely defined, we say that it is a family of algorithms, one per possible tie-breaking rule.

## Optimal Efficiency of A$^*$

The seminal work by Dechter and Pearl (1985) analyzes the optimal efficiency of A$^*$ in great depth, considering several degrees of optimal efficiency. They consider the heuristic as part of the input to the algorithm, so a problem instance is a tuple $\langle \Theta, h \rangle$. An instance is consistent if it has a consistent heuristic $h$. An algorithm is admissible if it is guaranteed to return an optimal plan for $\Theta$, whenever $h$ is admissible.

To prove optimal efficiency of an algorithm, some assumptions about the considered algorithms are needed. In their paper, Dechter and Pearl define a family of algorithms that use only a few primitive functions, such as expansion and heuristic functions. Eckerle et al. (2017) refine this by making explicit the assumption that all these functions are deterministic, and black box, defining the family of Deterministic, Expansion-based, Black Box (DXBB) algorithms. We also assume that the transition relation can only be accessed in a forward manner, as a function that given a state returns its successors. If backward search is possible, A$^*$ does not guarantee optimal efficiency (Chen et al. 2017).

**Definition 1** (*UDXBB* Algorithm). *A algorithm is Unidirectional, Deterministic, Expansion-based, Black Box (UDXBB) if it is deterministic and it has access to the state space $\Theta$ via exactly the following functions:*

- *Start: returns the initial state $s^I$.*

- *Is-goal: given a state $s$ returns true iff $s$ is a goal state.*

- *Expand: given a state $s$ returns a set of successor states $expand(s) = \{t \mid s \xrightarrow{l} t\}$.*

- *Cost: given a state and a successor state returns the cost of reaching it ($cost(s,t) = \min_{c(l)} s \xrightarrow{l} t$).*

They define a hierarchy with several degrees of optimality, based on comparing the sets of nodes expanded by different families of algorithms over a set of instances. Let $N(A, I)$ be the set of expanded nodes by algorithm $A$ on instance $I$. A family of algorithms $\mathcal{A}$ is X-optimally efficient over another $\mathcal{B}$ relative to an instance set $\mathcal{I}$ if:

- Type 0: $\forall I \in \mathcal{I}, \forall B \in \mathcal{B}, \forall A \in \mathcal{A}, N(A, I) \subseteq N(B, I)$.
- Type 1: $\forall I \in \mathcal{I}, \forall B \in \mathcal{B}, \exists A \in \mathcal{A}, N(A, I) \subseteq N(B, I)$.
- Type 2: $\forall I \in \mathcal{I}, \forall B \in \mathcal{B}, \forall A \in \mathcal{A}, N(B, I) \not\subset N(A, I)$.
- Type 3: $\forall B \in \mathcal{B}, \forall A \in \mathcal{A}, (\exists I_1 \in \mathcal{I}, N(A, I_1) \not\subseteq N(B, I_1)) \implies (\exists I_2 \in \mathcal{I}, N(B, I_2) \not\subseteq N(A, I_2))$

Among other results, Dechter and Pearl proved that, on consistent instances A$^*$ is 1-optimal, meaning that for any admissible $UDXBB$ algorithm $X$, there exists a tie-breaking for A$^*$ that expands a subset of the nodes expanded by $X$. They also show that no family of algorithms can be 0-optimal, meaning that there is no way to set the tie-breaking strategy to guarantee a minimal number of node expansions.

## Dominance Pruning

Dominance pruning is a technique that makes use of a dominance relation as an additional source of information. A relation $\preceq \subseteq S \times S$ is a dominance relation if, whenever $s \preceq t$, then $h_\epsilon^*(t) \le h_\epsilon^*(s)$. We say that a node $n_t$ prunes another $n_s$ if $n_s \ne n_t$, $g(n_t) \le g(n_s)$ and $s \preceq t$.

We define A$^*$ with dominance pruning (A$_{pr}^*$) as the vanilla A$^*$ algorithm with a simple modification. Anytime that a node $n_s$ is selected for expansion, skip it if there exists another node $n_t$ in open or closed such that $n_t$ prunes $n_s$. Nodes pruned this way are removed from open but they are neither expanded nor inserted into the closed list.[1] Therefore, pruned nodes are "forgotten" and no node can be pruned due to being dominated by a previously pruned node. This is necessary to correctly handle the case where there are only two nodes that prune each other, since in that case any of the two nodes could be pruned, but at least one of them must be expanded to find a solution.

In this work, we assume that the dominance relation is provided as an instance-dependent function. In practice, it can also be automatically obtained from a model of the problem, even though in this work we assume that the model is not available to the search algorithm. A common way to define a dominance relation is based on identifying resources (Hall et al. 2013), i.e. variables for which there exists a total order for their values such that larger values enable more actions. Furthermore, there are other more advance methods that find pre-orders on arbitrary abstract state spaces (Torralba and Hoffmann 2015). In both cases, the dominance relations that have been used in the literature are:

---

[1]Nodes can also be pruned upon generation to avoid the overhead of computing $h$ and open list insertion. But this does not affect the number of expanded states, which is what interests us.

- Pre-order relations: they are reflexive ($s \preceq s$ for all $s$) and transitive ($s \preceq t \wedge t \preceq u \implies s \preceq u$).

- Cost-simulation relations: whenever $s \preceq t$, for all $s \xrightarrow{l} s'$, either $s' \preceq t$ or $\exists t \xrightarrow{l'} t'$ s.t. $c(l') \leq c(l)$ and $s' \preceq t'$.

Even though one can define dominance relations that do not satisfy these properties, they are naturally obtained in most cases. In particular, the property of cost-simulation is related to the way automatic methods prove that the obtained relation is a dominance relation without having access to $h^*$.

## Definition of Optimal Efficiency

Following Dechter and Pearl, we are interested in the optimal efficiency of algorithms in regards of node expansions on concrete families of instances. In this section, we generalize their framework by considering the additional information of a dominance relation. This requires defining what consistent instances are in this case, as well as defining the different notions of optimal efficiency, and the families of algorithms that we will consider.

### Consistent Instances

A problem instance is a tuple $\langle \Theta, h, \preceq \rangle$, where $\Theta$ is a transition system, $h$ is an admissible heuristic for $\Theta$, and $\preceq$ is a dominance relation for $\Theta$. We say that an instance is consistent if both the heuristic and dominance relation are consistent on their own, and they are consistent with each other, meaning that they fulfill the following properties.

**Definition 2.** *An instance $I = \langle \Theta, h, \preceq \rangle$ is consistent if:*

*(i)* $h$ *is consistent.*
*(ii)* $\preceq$ *is a transitive cost-simulation.*
*(iii)* $\preceq$ *is consistent with $h$: $s \preceq t \implies h(t) \leq h(s)$.*

Condition (ii) ensures that the information provided by $\preceq$ is consistent in two different ways. First, $\preceq$ must be transitive, because if we do know that $s \preceq t$ and $t \preceq u$, then $h^*(u) \leq h^*(t) \leq h^*(s)$ so $s \preceq u$ can be deduced. Second, for a dominance relation to be consistent, we require it to be a cost-simulation relation so that whenever $n_t$ prunes $n_s$, then if $n_s$ or any of its successors would prune $n_u$, then $n_t$ or some of its successors prune $n_u$ as well.

Condition (iii) requires $\preceq$ and $h$ to not contradict each other on their comparison for any two states $s$ and $t$. Note that this does not render $\preceq$ uninformative, since comparing states based on their heuristic value is no substitute for dominance analysis. In particular, even if $\preceq$ always agrees with $h$, its role is to identify cases where the relative heuristic evaluation of both states is provably correct.

A question is whether these conditions are extremely rare or they can be expected to happen in practice. The first two conditions are indeed quite common: most heuristics that come from an optimal solution to a relaxation of the problem are indeed consistent; and typical approaches to compute dominance relations in planning are guaranteed to return transitive cost-simulation relations (Torralba and Hoffmann 2015; Torralba 2017).

An analysis of whether heuristics are consistent with respect to a dominance relation $\preceq$ is beyond the scope of this paper since that would require to consider concrete heuristic functions and dominance relations. In practice, it is reasonable to expect that most consistent heuristics will fulfill this property. For example, consider resource-based dominance relations that identify that states having more resources (fuel or money for example) are preferred. These are dominance relations because more resources can only enable more transitions in the state space; so heuristics that result from systematic (symmetric) relaxations of the problem will typically associate a lower heuristic value to states with more resources, everything else being equal. Indeed, for several families of heuristic functions in domain-independent planning, they have been shown to be consistent with symmetry equivalence relations (Shleyfman et al. 2015; Sievers et al. 2015), which are a special case of dominance relation. We conjecture that this holds as well for dominance relations based on comparing the values of sub-sets of variables, for heuristics that take into account the same subsets (e.g. we conjecture $h^{max}$ and $h^+$ are consistent with dominance relations over single variables, and pattern databases are consistent with dominance relations over subsets of the pattern).

### Types of Optimality

We extend the optimality criteria considered by Dechter and Pearl in several ways.

**Definition 3** (#-optimally efficient). *Let $N(A, I)$ be the set of expanded nodes by algorithm $A$ on instance $I$. A family of algorithms $\mathcal{A}$ is #-optimally efficient over another $\mathcal{B}$ relative to an instance set $\mathcal{I}$ if for any algorithm $B \in \mathcal{B}$ and instance $I \in \mathcal{I}$, there exists $A \in \mathcal{A}$ such that $|N(A, I)| \leq |N(B, I)|$.*

This definition of #-optimality is a relaxed variant of the 1-optimality definition by Dechter and Pearl, which requires the number of expansions by $A$ to be lower or equal than that of $B$, instead of requiring it to be a subset ($N(A, I) \subseteq N(B, I)$). Our criteria is slightly weaker since it only requires having an overall minimum number of expansions, which implicitly assumes that all expansions are equally time consuming. We say that 1-optimality is strict if $\mathcal{A}$ is 1-optimally efficient over $\mathcal{B}$, but $\mathcal{B}$ is not over $\mathcal{A}$. We say that #-optimality is strict if $\mathcal{A}$ is #-optimally efficient over $\mathcal{B}$, but $\mathcal{B}$ is not over $\mathcal{A}$.

We also consider when A$^*$ is optimal up to the last layer, i.e., where only nodes with an $f$-value lower than the optimal solution cost are taken into account. That is, we replace $N(X, I)$ by $N'(X, I)$ where $N'(X, I) = \{n \in N(X, I) \mid f(n) < F^*\}$. This is related to the notion of non-pathological instances introduced by Dechter and Pearl, which are those instances where A$^*$ does not expand any node $n$ with $f(n) = F^*$. However, paradoxically, non-pathological instances are very unlikely to occur in practice. For that reason, on the context of A$^*$ algorithms, we prefer to directly consider optimality up to the last layer, simply ignoring the effort that A$^*$ will make in the last $f$-layer, which most of the times strongly depends on the tie-breaking.

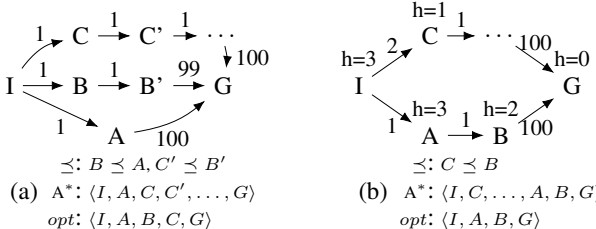

Figure 1: Summary of optimal efficiency relationships. All results assume consistent instances.

## Families of Algorithms

We introduce a new family of algorithms that extends *UDXBB* with dominance pruning.

**Definition 4** (*UDXBB_{pr}*)**.** *UDXBB_{pr} is a family of algorithms that extends UDXBB with the ability to perform dominance pruning, i.e., to discard any node $n_s$ if another node $n_t$ has been generated such that $n_t$ prunes $n_s$.*

Note that $UDXBB_{pr}$ algorithms cannot access the dominance relation directly or indirectly, i.e., they are not allowed to perform inference based on the fact that $h^*(t) \leq h^*(s)$ whenever $s \preceq t$. Our analysis focuses on using dominance pruning, excluding other future uses of dominance relations.

**Proposition 1.** *UDXBB_{pr} is strictly 1-optimal over UDXBB on all instances.*

*Proof.* 1-optimality follows trivially from the fact that $UDXBB$ is contained in $UDXBB_{pr}$, since $UDXBB_{pr}$ algorithms could choose not to prune any node if they desire so. To show this to be strict, it suffices to show an instance where there are nodes $n_s, n_t$ with $f(n_t) \leq f(n_s) \leq F^*$ such that $n_t$ prunes $n_s$. It is very easy to construct such example, e.g. see the instances in Figures 2, and 3. □

## Optimal Efficiency of $A_{pr}^*$

Thorough the paper, we assume consistent instances, i.e., that the heuristic function and dominance relation are consistent. Figure 1 summarizes our results. Our theoretical analysis concludes that, in terms of node expansions using dominance pruning is strictly better than not using dominance. Our main result is that, on consistent instances, the expansion order of $A^*$ is #-optimally efficient, meaning that there exist some tie-breaking of $A^*$ that expands a minimum number of expansions. We begin by showing some counter-examples on instances that do not satisfy our consistency criteria to highlight why consistency is required. Then we discuss how to characterize the states that must be expanded to find a solution and prove it to be optimal; prove our main result; and discuss what tie-breaking strategies are more appropriate for $A^*$ with dominance pruning.

## Counter-examples due to Inconsistencies

We begin considering the two things that characterize $A_{pr}^*$ algorithms from the set of $UDXBB_{pr}$ algorithms, and that may cause $A_{pr}^*$ to not be optimally efficient in inconsistent instances:

Figure 2: Counterexamples that show cases where $A_{pr}^*$ is not optimally efficient, when pruning according to the dominance relation below each figure. The "..." region represents an arbitrarily large region of the state space that will be expanded by $A_{pr}^*$, but could be avoided with a different pruning or expansion order strategies. In (a) $h = 0$ for all states, in (b) each node is labeled with its $h$ value.

1. A node is pruned whenever possible, and sometimes not pruning a node may lead to less overall expansions.

2. The expansion order of $A^*$ may not be optimally efficient anymore when considering dominance pruning.

Figure 2 shows examples where $A_{pr}^*$ does expand more nodes than necessary for these two reasons. The example in Figure 2a illustrates a state space and dominance relation $\preceq$ for which pruning a node whenever possible is not an optimal strategy, independently of the expansion order (for simplicity we set $h = 0$ for all states). After expanding the initial state $I$, one can prune node $B$ because it is dominated by $A$. However, if $B$ is pruned, $B'$ won't be generated under any expansion order so $C'$ and all its arbitrarily many successors will be expanded.

Our second example, illustrated in Figure 2b, shows a case where it is good to prune nodes whenever possible but the expansion order of $A^*$ leads to a sub-optimal number of expansions. The optimal expansion order is $\langle I, A, B, G \rangle$. $C$ does not need to be expanded even though $f(C) < F^*$ because $C$ will be pruned ($C \preceq B$ and $g(B) \leq g(C)$). However, $A_{pr}^*$ will expand $C$ after expanding the initial state $I$, since $f(C) < f(A)$ and $B$ has not been generated yet.

All these scenarios can be attributed to "inconsistencies" within the dominance relation $\preceq$ or between $\preceq$ and the heuristic function $h$. In Figure 2b the dominance relation and heuristic do not agree on the comparison between $B$ and $C$. The dominance relation proves that $B$ is at least as close to the goal as $C$, but the heuristic function estimates that $C$ is closer to the goal. In the case of Figure 2a, the dominance relation is inconsistent because the information that $A$ is closer to the goal than $B$ is lost after one expansion and neither $A$ nor any of its successors could be used to prune $B'$ or $C'$.

## Solution Sets

We first identify which states need to be expanded to prove optimality by any search that does not have access to any additional information, other than a heuristic function $h$ and the ability to prune nodes. Traditionally, this is done by identifying *must-expand* states that must be expanded for every algorithm to prove optimality, or *must-expand* pairs as done

in the bidirectional search setting (Eckerle et al. 2017). However, in our case there are many choices that can be made for dominance pruning, so now the difference between must-expand nodes and the nodes that belong to any concrete solution is not restricted to the last $f$-layer.

We define instead solution sets, which take into consideration all nodes that must be expanded by any $UDXBB_{pr}$ algorithm to find a solution, including the last $f$-layer. Let $\mathcal{S}$ be a set of nodes. We use $[\mathcal{S}]$ to denote the set extended with its immediate successors, i.e., $[\mathcal{S}] = \mathcal{S} \cup \{n_{s'} \mid n_s \to N_{s'}, n_s \in \mathcal{S}\}$. The intuition is that, if $\mathcal{S}$ is the set of nodes that have been expanded at some point during the execution of a $UDXBB$ algorithm, then $[\mathcal{S}]$ is the set of nodes that have been generated. In other words, if $\mathcal{S}$ represents the contents of the closed list, then $[\mathcal{S}] \setminus \mathcal{S}$ contains the set of nodes in the open list and all pruned nodes.

**Definition 5** ($UDXBB_{pr}$ Solution Set). *A set of nodes $\mathcal{S}$ is a $UDXBB_{pr}$ solution set for an instance $I$ if:*

*(a)* $\forall n_s \in \mathcal{S} \setminus \{n_{s^I}\}, \exists n_t \in \mathcal{S}, n_t \xrightarrow{l} n_s.$

*(b)* $\exists n_s \in [\mathcal{S}], s \in S^G$ and $g(n_s) = F^*,$

*(c)* $\forall n_s \in [\mathcal{S}] \setminus \mathcal{S}, f(n_s) \geq F^*$ or $\exists n_t \in \mathcal{S}, n_t$ prunes $n_s.$

Condition (a) requires that every expanded node in $\mathcal{S}$ was generated by expanding one of its parents. Condition (b) requires that an optimal solution was found. Condition (c) ensures that the solution found is proven to be optimal, because all nodes in the open list after expanding $\mathcal{S}$ have a large enough $f$-value or are pruned by dominance.

**Theorem 1.** *Let $I$ be any admissible instance. Then, expanding a solution set is a necessary and sufficient condition for admissible $UDXBB_{pr}$ algorithms, i.e., for any $A$ in $UDXBB_{pr}$, $N(A, I)$ is a solution set.*

*Proof Sketch.* If (a), (b), and (c) hold, then an optimal solution has been found due to (a) and (b), and the stopping condition holds, since all states remaining in the open list have an $f$-value greater or equal to the incumbent solution.

If (a) does not hold, then a state has been expanded without being in open, which is impossible in any $UDXBB$ algorithm. If (b) does not hold, then an optimal solution has not been found. If (c) does not hold, then there exists some $s$ in the open list that may lead to a solution cost lower than $F^*$, so the solution was not proven to be optimal. $\square$

We remark that the proof above relies on the fact that $UDXBB_{pr}$ algorithms are not allowed to use dominance relations for anything except dominance pruning. Otherwise, the criteria (c) of a solution set could be made weaker, increasing the set of possible solution sets.

## $A^*_{pr}$ is Optimally Efficient on Consistent Instances

Before proving our main result of this section, we analyze some properties that hold for consistent instances. An important one is that, whenever $h$ and $\preceq$ are consistent with each other, nodes with larger $f$-value cannot prune nodes with lower $f$-value.

**Lemma 1.** *Let $I$ be a consistent instance. Let $n_s, n_t$ be any two nodes such that $n_t$ prunes $n_s$. Then, $f(n_t) \leq f(n_s)$.*

*Proof.* Since $n_t$ prunes $n_s$, it holds that $g(n_t) \leq g(n_s)$ and $s \preceq t$. By consistency, $h(t) \leq h(s)$, so $f(n_t) \leq f(n_s)$. $\square$

Next, we show that pruning is transitive.

**Lemma 2.** *If $\preceq$ is transitive, $n_u$ prunes $n_t$ and $n_t$ prunes $n_s$, then $n_u$ prunes $n_s$.*

*Proof.* By the assumption it follows that $g(n_u) \leq g(n_t) \leq g(n_s)$, and $s \preceq t \preceq u$. Therefore, $g(n_u) \leq g(n_s)$ and, by transitivity of $\preceq$, $s \preceq u$. So $n_u$ prunes $n_s$. $\square$

Next, we show that all states in the smallest solution set must be expanded only with its optimal $g$-value.

**Lemma 3.** *Let $I$ be a consistent instance. Then, there exists a solution set $\mathcal{S}$ for $I$ of minimum size such that for all $n_s \in \mathcal{S}$, $g(n_s) = g^*(s)$.*

*Proof.* Assume the contrary. Then, some $n_s$ has been expanded with a sub-optimal value, $g^*(s) < g(n_s)$. Therefore, a predecessor along the optimal path from $s^I$ to $s$ has not been expanded. Let $n_t$ be the first such predecessor. By consistency of $h$, we know that $f$-values monotonically increase along a path, so $f(n_t) \leq f^*(n_s) < f(n_s)$. As $n_t \notin \mathcal{S}$, by condition (c) of a solution set, $n_t$ was pruned, i.e., $\exists n_u \in \mathcal{S}$ s.t. $n_u$ prunes $n_t$. As $\preceq$ is a cost-simulation, then $n_u$ has some successor that would prune $n_s$, so there must be a node in $\mathcal{S}$ that prunes $n_s$. Therefore, $\mathcal{S} \setminus \{n_s\}$ is also a solution set, contradicting the fact that $\mathcal{S}$ is of minimum size. $\square$

We next show that pruning a node whenever possible is an optimally efficient strategy because there exists a solution set $\mathcal{S}$ of minimum size that does not contain any node that can be pruned by another node in $[\mathcal{S}]$, unless both nodes prune each other. To show this, we consider Algorithm 1.

---

**Algorithm 1:** Replace

**Input:** $\mathcal{S}_0, n_s \in \mathcal{S}_0, n_t \in [\mathcal{S}_0]$ where $\mathcal{S}_0$ is a solution set and $n_t$ prunes $n_s$

**Output:** Solution set $\mathcal{S}_i$ that does not contain $n_s$

1   $\mathcal{S}_1 := (\mathcal{S}_0 \cup \{n_t\}) \setminus \{n_s\}$;

2   $i = 1$;

3   **while** $\exists n_{s^i} \in \mathcal{S}_i, \nexists n_{u^i} \in \mathcal{S}_i, n_{u^i} \xrightarrow{l} n_{s^i}$ **do**

4      Choose such an $n_{s^i}$ with minimum $g$-value ;

5      Choose $n_{t^i}$ in $[\mathcal{S}_i]$ such that $n_{t^i}$ prunes $n_{s^i}$ ;

6      $\mathcal{S}_{i+1} := \mathcal{S}_i \cup \{n_{t^i}\} \setminus \{n_{s^i}\}$ ;

7      $i = i + 1$ ;

8   **return** $\mathcal{S}_i$ ;

---

**Lemma 4.** *Let $\mathcal{S}_0$ be a solution set for a consistent instance. Let $n_s \in \mathcal{S}_0$, $n_t \in [\mathcal{S}_0]$ such that $n_t$ prunes $n_s$. Then, Algorithm 1 returns a solution set $\mathcal{S}_k$ such that: $|\mathcal{S}_k| \leq |\mathcal{S}_0|$; $n_s \notin \mathcal{S}_k$; $n_t \in \mathcal{S}_k$; and If $n_t \in \mathcal{S}_0$ then $|\mathcal{S}_k| < |\mathcal{S}_0|$.*

*Proof Sketch.* The size of the solution set cannot increase during the execution of Algorithm 1, i.e., $|\mathcal{S}_{i+1}| \leq |\mathcal{S}_i|$ because a node is removed at each iteration and at most one node is added. If $n_t \in \mathcal{S}_0$ then $|\mathcal{S}_1| = |\mathcal{S}_0| - 1$, since

$n_s$ was removed and no node was added, so in that case $|\mathcal{S}_k| \leq |\mathcal{S}_1| < |\mathcal{S}_0|$.

Properties (b) and (c) of a solution set are preserved by all intermediate $\mathcal{S}_i$ because $n_{s^i}$ is replaced by $n_{t^i}$ such that $n_{t^i}$ prunes $n_{s^i}$, so by Lemma 1 and 2, $n_{t^i}$ can do anything $n_{s^i}$ could. Property (a) holds when the algorithm terminates, since it is the stopping condition for the loop. The algorithm always terminates because all nodes $n_{s^i}$ removed in the loop are descendants of $n_s$ which were present in $\mathcal{S}_0$, and there are only finitely many.

Finally, it remains to be proven that, at every iteration in line 4, there exists some $n_{t^i}$ in $[\mathcal{S}_i]$ such that $n_{t^i}$ prunes $n_{s^i}$. Note that $n_{s^i}$ is a descendant of $n_s$ that has no parent in $\mathcal{S}_i$. Since all nodes in $\mathcal{S}_0$ have a parent, and all $n_{t^i}$ added along the way too, this means that the parent of $n_{s^i}$ was some $n_{s^j}$ removed in a previous iteration $j < i$, being replaced by $n_{t^j}$. Since $\preceq$ is a cost-simulation relation, $n_{t^j}$ must have a successor $n_{t^i}$ that prunes $n_{s^i}$. $\square$

**Lemma 5.** *Let $\mathcal{S}$ be a solution set of minimum size for a consistent instance. Then there do not exist $n_s, n_t$ in $\mathcal{S}$ such that $n_t$ prunes $n_s$.*

*Proof.* Assume that $n_t$ prunes $n_s$. By Lemma 4, using the procedure above, we can construct another solution set $\mathcal{S}'$ such that $|\mathcal{S}'| < |\mathcal{S}|$, contradicting that $\mathcal{S}$ has minimum size. $\square$

**Lemma 6.** *Let $I$ be a consistent instance. Then, there exists a solution set $\mathcal{S}$ of minimum size for $I$ such that there does not exist any $n_s \in \mathcal{S}$ and $n_t \in [\mathcal{S}]$ such that $n_t$ prunes $n_s$ and $n_s$ does not prune $n_t$.*

*Proof Sketch.* Assume the opposite, let $\mathcal{S}$ be a solution set such that there exist $n_s \in \mathcal{S}$ and $n_t \in [\mathcal{S}]$ where $n_s$ prunes $n_t$ and $n_t$ does not prune $n_s$. By Lemma 5 $n_t \notin \mathcal{S}$. By condition (c) of a solution set we know that either $f(n_t) \geq F^*$ or there exists $n_u \in \mathcal{S}$ such that $n_u$ prunes $n_t$.

Case 1: There exists $n_u \in \mathcal{S}$ such that $n_u$ prunes $n_t$. By transitivity, $n_u$ prunes $n_s$, so one can construct a minimal solution set with Lemma 4 of smaller size, contradicting that $\mathcal{S}$ is a solution set of minimal size.

Case 2: $f(n_t) \geq F^*$. Then, by Lemma 1, $f(n_s) \geq f(n_t) \geq F^*$. If $f(n_t) > F^*$, we can remove $n_s$ and all its descendants from $\mathcal{S}_0$ to obtain a smaller solution set, contradicting the fact that it is a solution set of minimal size. Therefore, $f(n_s) = f(n_t) = F^*$. Note that a solution set of minimum size only contains a node with $f(n_s) = F^*$ when $n_s$ is on the solution path returned by the algorithm. This path can be replaced by another of the same length and cost that goes through $n_t$ by repeatedly calling Algorithm 1. $\square$

Now we are ready to prove our main result.

**Theorem 2.** *$A^*_{pr}$ is #-optimal on consistent instances over $UDXBB_{pr}$.*

*Proof.* We show that there exists a solution set $\mathcal{S}$ of minimum size for which there exists a tie-breaking strategy under which $A^*_{pr}$ with $h$ and $\preceq$ expands exactly $\mathcal{S}$. By Lemma 6, we choose $\mathcal{S}$ so that there does not exist any $n_s \in \mathcal{S}$ and $n_t \in [\mathcal{S}]$ s.t. $n_t$ prunes $n_s$ and $n_s$ does not prune $n_t$. Assume

a tie-breaking that prefers expanding nodes in $\mathcal{S}$ over any other node, and prefers pruning nodes not in $\mathcal{S}$. Formally, our tie-breaking strategy selects for expansion any node not in $\mathcal{S}$ such that it can be pruned. If no such node exists, it selects a node (with minimal $f$ value) from $\mathcal{S}$ that cannot be pruned. This tie-breaking always succeeds because otherwise, the open list does not contain any node with minimal $f$ value that is outside $\mathcal{S}$ and can be pruned or that it is in $\mathcal{S}$ and cannot be pruned. Then, the node selected for expansion either: (A) it is in $\mathcal{S}$ but can be pruned due to some node in open or closed; (B) it is not in $\mathcal{S}$ and cannot be pruned.

Case (A). There exists $n_t$ that prunes some $n_s \in \mathcal{S}$. By Lemma 5, we know that $n_t \notin \mathcal{S}$. As $n_t$ is in the open list after having expanded a subset of $\mathcal{S}$, $n_t \in [\mathcal{S}]$ and, by our choice of solution set with Lemma 6, $n_s$ prunes $n_t$. By Lemma 1, $f(n_t) \leq f(n_s)$, so with our tie-breaking strategy $A^*$ would have selected $n_t$ instead, reaching a contradiction.

Case (B). Let $n_s$ be a node that is expanded by $A^*_{pr}$ but it is not in $\mathcal{S}$. If $f(n_s) = F^*$, then a node along the optimal solution contained in $\mathcal{S}$ should have been chosen instead. If $f(n_s) < F^*$, by condition (c) of a solution set, there exists $n_t \in \mathcal{S}$ such that $n_t$ prunes $n_s$. Again, if $n_t$ is in open or closed, $n_s$ will be pruned reaching a contradiction. Otherwise, there must be an ancestor along the path from $s^I$ to $n_t$ in open with its optimal $g$-value. Such an $n_u \in \mathcal{S}$, must have $f(n_u) \leq f(n_t) \leq f(n_s)$, so according to our tie-breaking $n_u$ would have been chosen for expansion instead of $n_s$ ($n_u$ cannot be pruned by the same argument as in case (A)). $\square$

**Corollary 1.** *$A^*_{pr}$ is strictly #-optimal over $A^*$ on consistent instances.*

*Proof.* This follows directly from the fact that $A^*_{pr}$ is #-optimal over $UDXBB_{pr}$ and $UDXBB_{pr}$ is strictly 1-optimal over the family of $UDXBB$ algorithms, which contains all algorithms in the family of $A^*$ algorithms. $\square$

## Optimal Tie-Breaking Strategies

For $A^*$ with consistent heuristics the tie-breaking strategy is only relevant in the last $f$-layer. Ideally, once the minimum $f$-value in the open list is equal to $F^*$, only nodes on a path to the goal will be selected for expansion. Practical implementations often prefer expanding nodes with lowest $h$-value, aiming to reduce the effort in the last layer. In domain-independent planning, where a factored model of the state space is available to offer additional information to the algorithm, some other strategies have been suggested, like using (possibly inadmissible) heuristic functions that estimate plan length instead of plan cost (Asai and Fukunaga 2017; Corrêa, Pereira, and Ritt 2018). They showed that tie-breaking can be quite significant for the overall performance, specially in domains with 0-cost actions.

$A^*_{pr}$, however, is more sensitive to the choice of tie-breaking strategy, since it may matter along previous layers. This brings up the question of what is a good tie-breaking strategy for $A^*_{pr}$. We define $A^*_{g<,pr}$ as $A^*_{pr}$ breaking ties in favor of states with minimum $g$-value.

**Theorem 3.** *$A^*_{g<,pr}$ is 1-optimal efficient up to the last layer over $A^*_{pr}$ on consistent instances.*

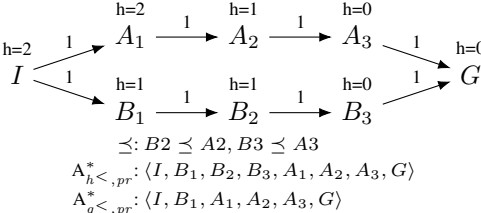

$\preceq$: $B2 \preceq A2$, $B3 \preceq A3$

$A^*_{h<,pr}$: $\langle I, B_1, B_2, B_3, A_1, A_2, A_3, G \rangle$

$A^*_{g<,pr}$: $\langle I, B_1, A_1, A_2, A_3, G \rangle$

Figure 3: Counter-example for the 1-optimal efficiency of $A^*_{h<,pr}$ up to the last layer on consistent instances.

*Proof Sketch.* Let $\mathcal{S}$ to be a solution set for $A^*_{pr}$, and let $\mathcal{S}'$ be the subset of nodes in solution set up to the last layer, $\mathcal{S}' = \{n_s \in \mathcal{S} \mid f(n_s) < F^*\}$. We show that there is an expansion order compatible with $A^*_{g<,pr}$ that expands all nodes in $\mathcal{S}'$ before expanding any other node. For this, the same proof from Theorem 2 applies up to case (B). For case (B), we know that $f(n_s) < F^*$ and, by the same argument as in the proof of Theorem 2, some $n_u \in \mathcal{S}$ must remain in the open list with $f(n_u) \leq f(n_t) \leq f(n_s)$. At this point the tie-breaking matters since whenever $f(n_u) = f(n_s)$, the tie-breaking policy should allow selecting $n_u$ over $n_s$. Since $n_u$ is an ancestor of $n_t$, $g(n_u) \leq g(n_t)$, and since $n_t$ prunes $n_s$, $g(n_u) \leq g(n_t) \leq g(n_s)$. Then, expanding $n_u$ instead of $n_s$ is still valid according to the $g^<$ tie-breaking strategy. $\square$

However, the same is not true for every tie-breaking strategy for $A^*$. For example, let $A^*_{h<,pr}$ be the family of $A^*_{pr}$ algorithms with a tie-breaking strategy that always prefers a state with minimum $h$-value. As argued above this is the tie-breaking preferred by most implementations of $A^*$ without dominance pruning, but it cannot guarantee anymore that the number of expansions up to the last layer will be minimal.

**Theorem 4.** *$A^*_{h<,pr}$ is not optimally efficient up to the last layer on consistent instances.*

*Proof Sketch.* Figure 3 shows a counter-example of a consistent instance where all tie-breaking strategies compatible with $A^*_{h<,pr}$ expand a node that $A^*_{g<,pr}$ would not expand. After expanding $I$ and $B_1$, the open list contains two nodes: $B_2$ and $A_1$, both with an $f$-value of 3. At this point, $A_2$ has not been generated yet so $B_2$ cannot be pruned. However, $A^*_{h<,pr}$ will expand $B_2$ and $B_3$ (and in general the entire plateau of states with $f = 3$ underneath $B_2$), before expanding $A_1$. Note that this happens for nodes with $f = 3 < 4 = F^*$, i.e. nodes before the last $f$-layer. $\square$

**Corollary 2.** *$A^*_{g<,pr}$ is strictly 1-optimally efficient up to the last layer over $A^*_{h<,pr}$ on consistent instances.*

*Proof Sketch.* 1-optimality follows directly from Theorem 3, since $A^*_{h<,pr}$ is contained in $A^*_{pr}$. The fact that optimality is strict follows from Theorem 4. $\square$

Thus, there are two conflicting objectives. Up to the last layer, it is provably beneficial to break ties in favor of lower $g$-value. On the last layer, empirical analysis show that it is better to break ties in favor of lower $h$-value. Which one should be given priority depends on the particular domain, dominance relation and heuristic. Our preliminar experiments show that in common planning domains, it is often beneficial to break ties in favor of lower $h$-value even when dominance pruning is used.

## Conclusions

We analyzed the optimal efficiency of $A^*$ with dominance pruning, $A^*_{pr}$. Assuming a consistent heuristic is not sufficient, because there may be inconsistencies in the dominance relation as well, which may cause $A^*_{pr}$ to perform unnecessary expansions. We defined a new criterion of consistency for heuristic and dominance relation pairs, which ensures that $A^*_{pr}$ will be optimally efficient in terms of the number of expanded nodes. We also show that tie-breaking in favor of nodes with lower $g$ value is provably preferable to minimize the number of expansions up to the last layer. This contrasts with common strategies, which favor nodes with lowest $h$-value to minimize expansions in the last layer.

As in the optimal efficiency result for $A^*$, our analysis is based only on the number of state expansions and it ignores the actual runtime. There are of course other algorithms which may outperform $A^*$ according to different performance measures. For example, the IDA$^*$ algorithm (Korf 1985) and other extensions like Budgeted Tree Search (Helmert et al. 2019; Sturtevant and Helmert 2020) outperform $A^*$ in terms of memory usage. EPEA$^*$ (Goldenberg et al. 2014) aims to minimize the number of nodes generated, which is arguably more relevant to runtime than expanded nodes, but it requires additional domain-specific knowledge. Finally, other algorithms may outperform $A^*$ in terms of runtime, e.g., when the benefits of reducing the number of node expansions does not compensate the overhead of computing the heuristic or performing pruning, which may require a quadratic cost in the number of generated states in the worst case. Nevertheless, for concrete problems and/or dominance relations it may be possible to perform the pruning more efficiently (e.g., dividing states in classes so that each state needs to be compared only against a small subset of alternatives), and one could extend rational algorithms that reason about when it is worth to compute the heuristic (Barley, Franco, and Riddle 2014; Karpas et al. 2018) to consider dominance as well.

Finally, in this work we extended the basic framework with the ability of dominance pruning using a dominance relation, but it could also be extended in other ways. For example, if backward search is possible, there are a variety of bidirectional heuristic search algorithms that can outperform $A^*$ in terms of node expansions (Eckerle et al. 2017; Chen et al. 2017). One could consider several extensions of this paradigm regarding different forms of dominance, e.g., introducing variants that make use of more general forms of dominance (Torralba 2017), or alternative methods to exploit this information. This may open new avenues of research on how to use dominance relations beyond dominance pruning in order to make the most of them.

## Acknowledgments

Álvaro Torralba was employed by Saarland University and the CISPA Helmholtz Center for Information Security during part of the development of this paper. I would like to thank Nathan Stutervant for his insights on DXBB algorithms. Thanks to the anonymous reviewers at SoCS'20 and HSDIP'20 as well as to the Basel reading group for their comments that helped me to improve the paper.

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
