# OpenReview forum: "On the Optimal Efficiency of A* with Dominance Pruning"
_icaps-conference.org/ICAPS/2020/Workshop/HSDIP — HSDIP 2020_

### Official Review · AnonReviewer2 · 2020-03-30
**Early assessment of submission "On the Optimal Efficiency of A* with Dominance Pruning"**

**Rating:** 8
**Confidence:** 4

**Review:**

This short paper presents a theoretical evaluation of optimality results regarding A* with dominance pruning and fits completely in the scope of the HSDIP workshop. I do not have many comments or questions for my early assessment; the paper is clearly written, as far as I can see formally correct, and would provide a good addition to the workshop.

Since we do not have a short paper page limit for HSDIP I would like to ask the authors to consider adding a brief comment on how dominance relations are generated in general. The methods at the end of paragraph "Dominance Pruning" require that some dominance relation already exists, how does one get an initial dominance relation that can get extended?

It took me a while to understand the third example of Figure 1, and I think the biggest issue was the phrase "one can use h(B') instead of h(C)" because it is not completely clear to what 'use' refers. Apparently it is to expand states (compute the f-value), but the part before this sentence is about pruning nodes.

Before Definition 3 [S] is introduced as a set of states. But in the definition it is then a set of nodes (or the subscript of n_s \in [S] is wrong).

Some inline definitions or equations should have an additional space (\,), e.g. "A relation $\succ \, \subseteq S \times S$ is a dominance relation" or in equation (b) and (c) of Definition 3.

Grammar / other:

- "while retaining any _optimally_ guarantees"
- "lead to a _globally_ number of expansions"
- "work by Dechter and Pearl analyze the optimal efficiency" => analyzes
- "We define A* with dominance pruning as the vanilla A* algorithm with a simple modification of A*" =>last mention of A* is unnecessary
- "Current methods for finding dominance relations obtain relations are:"
- "The optimal expansion order is I, A ,B, B',G" => in (b) there is no B'
- "In the third example shows an example"
- "We denote [S] to set extended with its immediate successors as , i.e."
- Proof of Theorem 2: "By Lemma , we choose"
- "Then, the state selected for expansion either: (A) it is in S [...]" => remove 'it'

Question:

It seems that the next step would be to show for currently used heuristics/dominance relations that requirement (iii) of Definition 2 is satisfied. You write that showing this is out of scope for the paper. How hard do you think will it be to show this? If we can show this, what would it mean for systems that already apply dominance pruning?

---

### Official Review · AnonReviewer1 · 2020-03-31
**Early assessment**

**Rating:** 8
**Confidence:** 4

**Review:**

The paper analyzes under which circumstances A* with dominance pruning is optimally efficient. It is written mostly clearly and I could not find any faults. Overall a very interesting paper!

I do however have some questions where I think the paper could be clearer:

1) It took me a while to realize nodes whose states dominate each other and have equal g-value are not pruning each other (the first node is pruned since the second node is in open, but then for the second node the first node neither in open nor closed, correct?). Granted it is a small detail but I think putting this explanation in a footnote would be helpful.

2) The first sentence of "Counter-examples due to Inconsistencies" reads to me as if those three examples are the only ones that can happen, since you say "we consider *the* three reasons". Is this your intention? If so, why can there be no other reasons? If not, I would suggest to simply say "we consider three reasons".

3) I first assumed that your examples want to show that A* with dominance pruning might expand more nodes than A* without dominance pruning, but this is not the case, correct? Can this even happen? I would appreciate a clarification.

4) You assume unit costs, but your examples are not unit cost. Can these examples also happen with unit cost tasks?

5) It is not fully clear to me why you need [S]. Is it to get around awkwardness with the last f-layer?

6) You say you only consider unit costs to simplify the presentation, implying that the results hold for non-unit costs as well. Is this the case and if so why? It seems to me that for example Lemma 5 only works with unit costs.


Minor comments:

- introduction: "lead to a globally number of expansions" -> globally lower number?
- background: Two formulas (s->(l)t and h(s)-h(t) \leq c(l)) follow each other without text in between, which is a bit awkward to read.
- dominance pruning: "which interest*s* us"
- Figure 1: It was not immediately clear to me that the sets of tuples are dominance relations, and that (B,A) means A dominates B.
- The stated optimal expansion order of example 2 contains B', but there is no B' in Figure 1b.
- consistency of heuristics and dominance pruning: "A problem instance is..." - you defined problem instance before (but for A* without dominance pruning). Maybe write something like "In the setting we consider, a problem instance is..."
- "We denote [S] to set extended with..." -> to S extended with?
- You define F* as the cost of an optimal plan, but use f* afterwards
- Lemma 4 proof: "As n_t \not\in S." I do not know to what this sentence belongs.
- Lemma 4 proof: f(n_s) \leq F* and thus f(n_t) \leq F*, meaning n_t must have been pruned, right? I would add this explanation.
- Lemma 7: "prunes n_t* *and" (missing space)

---

### Author Response · Authors · 2020-07-23
**Thanks for the reviews**

Thanks to both reviewers for the early feedback and also to the anonymous SOCS'20 reviewers that provided a lot of feedback about the first version of this paper.

We've tried to fix all your comments and also extended the paper with additional results, so it's quite a different version. Let us summarize here the main changes:
* Reviewers asked us to clarify our assumptions about the class of algorithms being considered. We've tried to do so in this version, highlighting that we assume algorithms that do dominance pruning.
* We support costs in the current version of the paper, and not assume unit costs anymore.
* We rewrote several proofs to fix some minor issues in the previous version.
* We added new results comparing A* with dominance pruning against A*, and different tie-breaking strategies for A* with dominance pruning.

We also want to answer the comment by reviewer 2, which was also shared by one of the reviews at SoCS. We consider to analyze the consistency of heuristics and dominance relations out of the scope of this paper. The main reason for this is that doing this analysis would require to consider concrete dominance relations and concrete heuristics. This is of course domain dependent. One could do this for well-known families of relations and heuristics in planning, but this is clearly out of the scope of this paper: our results are applicable to any search space regardless of whether they come from a planning task.

Nevertheless, we have very good reasons to believe that most consistent heuristics will be consistent with dominance relations, and we have tried to incorporate them in the paper. Current dominance relations compare states by comparing individual variables one by one (or by considering projections onto several variables at once). For example, by noticing that if two states differ only on the amount of fuel of truck1, the one with more fuel is better.

A heuristic being inconsistent with this would assign lower heuristic values to states with less fuel, which is unlikely for any heuristic to do. In general, we conjecture that if the heuristic is consistent (e.g. h1, h+ or PDBs) and considers the same or larger projections than the dominance, then both will be consistent. We did not incorporate this into the paper because having a formal proof would require to introduce in detail planning tasks, several families of planning heuristics, and several families of relations.

---

### Comment · AnonReviewer2 · 2020-08-13

Brief summary of the paper:

The paper presents a theoretical evaluation of the node expansion behaviour of
A* search algorithms when some dominance relation allows the search to prune
nodes. The paper shows that if consistency guarantees between the dominance
relation and the heuristic are not met, then A* with dominance pruning
potentially expands more nodes than classical A* without any pruning.
Following that, the paper formally shows that if the consistency guarantees hold
then there exists a tie-breaking strategy for A* with dominance pruning that
expands strictly less nodes than any classical A* algorithm. Moreover,
the paper shows that it is beneficial if the tie-breaking strategy additionally
prefers states of shorter path length, while this is not true for tie-breaking
strategies that prefer states with lower heuristic estimates.

Summary of the review:

The paper got significantly extended to the previous version, is still very
clearly written and the topic of course still fits into the HSDIP workshop. I
found the paper very interesting; although it is a purely theoretical evaluation
it clearly shows the pitfalls of having a dominance relation that is
inconsistent with the underlying heuristic, but it also shows that having such
consistency guarantees can lead to strictly less expansions. The discussion on
the different tie-breaking strategies is also intriguing, and the results of the
paper warrant an even deeper look into existing dominance pruning strategies.

Major comments:

I assume the authors aim to send this paper to a major AI conference, e.g. AAAI,
so I want to give some comments / suggestions with that in mind.

- I agree with that the analysis of consistency between known heuristics and
dominance relations is out of scope of the paper. Personally I don't consider
the phrase "which seems unlikely" as a strong argument. I found the comment in
the rebuttal

> In general, we conjecture that if the heuristic is consistent (e.g. h1, h+
> or PDBs) and considers the same or larger projections than the dominance, then
> both will be consistent.

much more convincing, especially if you follow then with the example of symmetry
equivalence relations.

- I wonder whether it is possible to give the intuition of Lemma 4 / Algorithm 1
with a figure where a large circle denotes S, s being part of that circle, and
[S] is depicted by a dashed circle enclosing S, which includes t. Then s is
removed and the circle now represents S_1 which includes t. Additionally, the
you could depict s->s', which when s is removed symbolises that s' is some
state for which S_1 contains no parent, so s' has to be replaced by some state
t' with s' \prec t' as well (and since \prec is a cost simulation either t = t'
or t' is some successor of t).
All in all, the proof of Lemma 4 was probably the hardest to follow, not only
because its almost a whole column, but also because of the many different state
declarations (s^i, t^i, u, t',v',v,w,s^j, t^j).

---

> ### Comment · AnonReviewer2 · 2020-08-13
>
> (split in two parts because the whole review exceeds 5000 characters)
>
> Minor comments:
>
> - Definition 1: "A algorithm" -> "An algorithm"
> - "\mathcal{A} is X-optimally efficient over another \mathcal{B}" -> another
> family of algorithms \mathcal{b}
> - In the following paragraph it says "any admissible UDXBB algorithm X", but
> just before X is used to denote 'X-optimally'.
> - "Therefore, pruned nodes are "forgotten" and no other node can be pruned due
> to being dominated by another node.", I suggest to write "due to being dominated
> by the pruned node" rather than "due to being dominated by another node".
> - "Furthermore, there are more advance methods" -> advanced
> - "We replace N(X,I) by N'(X,I)", strictly speaking X is not defined
> - "Paradoxically, non-pathological instances are very unlikely to occur in
> practice" - can you give a source for that statement?
> - Figure 1: it should somewhere say that the direction of the arrow denotes the
> relation order, as that sometimes differs per research area, i.e. whether A -r> B
> means A r B, or B r A.
> - "It is very easy to construct such example" -> such an example
> - "Thorough the paper" -> Through the paper
> - "We begin considering the two things" -> We begin by considering. I also
> suggest to use a different word than 'things', e.g. 'features' or 'aspects' or
> 'traits'.
> - Figure 2: A^* should be A^*_pr
> - "that must be expanded for every algorithm to prove optimality" -> by any
> algorithm
> - "during the execution of a UDXBB algorithm" -> of an UDXBB algorithm
> - Proof of Lemma 3: "As n_t \notin S. By condition [...]", comma instead period?
> - Proof of Lemma 4: $f(n_{t'}) \leq f(n_{t^i}) \leq F^*$, the latter relation
> should probably be <, because we have $f(n_{t'}) < F^*$.
> - Proof of Lemma 4: "satisfy property (i)", do you mean property (a)?
> - Proof of Lemma 4: I suggest to write "there has to be n_v \in S_i such that
> n_v prunes n_{t^i}" instead of "there is n_v \in S_i" to emphasize that this
> follows from n_{t^i} not being in S_i.
> - Proof of Theorem 2, $n_s prunes n_t$ (top of the new page) is formatted
> incorrectly (probably should say $n_s$ prunes $n_t$)
> - Proof of Corollary 1: "all algorithms if the family of A* algorithms" ->
> of/from the family of A* algorithms
> - Proof of Theorem 3: "the subset of nodes in solution set" -> in the solution
> set
> - Theorem 4 says "not optimally efficient", but strictly speaking "not optimally
> efficient" is not defined (which 'X'-optimally is it?)
> - "Assuming a consistent heuristic is not sufficient, because there may be
> inconsistencies in the dominance relation as well, which could lead A* to expand
> more nodes than necessary" - grammar / missing word
> - "This contrast with common [...]" -> This contrasts
> - "More relevant to runtime that the number of nodes expanded" -> than the
> number of nodes
>
> - I am not sure if I agree with the naming 'n-optimally efficient'. The
> definitions by Dechter and Pearl also say 'k-optimally' (or Type-k optimal), but
> they talk about subset relations. I would say that 'n-optimal' implies a similar
> relationship (which is the case, because the relation is on number of
> expansions), but according to some parameter n, which is not the case. I guess
> 'n' stands for number, but given that n is often regarded as some integer
> variable (similar to k) I find that not a good choice. Why not simply say
> 'optimally expansion efficient', since some A guarantees to never expand more
> nodes than any B?
>
> - In the first half of the paper state successors are usually denoted with '
> (e.g.  s->s', A -> A') and dominance pairs are denoted as s and t (s \prec t).
> But in definition 5 a) we suddenly have t -> s instead of s -> s'. Then, in the
> proof of Lemma 3 we suddenly have t being a predecessor of s. Formally there is
> nothing wrong, but changing the intuition of s,s' and s,t that the reader has
> become familiar with by that time makes it sometimes hard to follow.
> I am also not sure why you use sometimes 'u' as predecessor instead of 'r'
> (alphabetically it is r,s,t).
>
>
> Questions:
>
> - In the proof of Corollary 2 it says that A*_h is contained in A*_g. Why does a
> the set of tie-breaking strategies which prefer a state with minimum g value
> include preferring tie-breaking strategies which prefer a state with minimum h value?
>
> - The tie-breaking strategy given in the proof of Theorem 2 requires to know the
> solution set S, which in reality is of course not known beforehand. Do you have
> some intuition on how likely it is that such a tie-breaking occurs in practice?

---

### Comment · AnonReviewer1 · 2020-08-14

The paper has been majorly rewritten since my initial assessment, but
my overall evaluation did not change (except for the better). It gives major
insights into theoretical aspects of dominance pruning, is correct as far as I
can tell, and is very clearly written, so a clear accept from my perspective.

I have however some additional comments over some details:

1. n-optimally efficient sounds to me like it belongs to the type hierarchy
discussed earlier in the paper. I assume n should symbolize "number of
expansions"? Maybe something like |N|-optimality would make this clearer?

2. In Lemma 6, should n_s and n_t not be reversed, i.e. n_s \in S and n_t
\not\in [S] such that n_t prunes n_s and n_s does not prune n_t? This also
applies to the first paragraph in the proof of Theorem 2.

3. Why is A*_{h<,pr} contained in A*_{g<,pr}? (proof sketch for Corollary 2)

Minor comments:
 - Introduction: "since states that are ..., are not necessarily better.." -> I
 would remove the comma
 - Last paragraph of Optimal Efficiency of A^*_{pr}: Then discuss how... -> Then
 we discuss how
 - Solution sets: S = closed list, [S] \ S = open list -> what about pruned
 states? They should be in [S] \ S but not in open, correct?
 - Proof for Lemma 3: "As n_t \not\in S". -> not a full sentence
 - Lemma 4: "the algorithm" -> Algorithm 1
 - Proof for Lemma 4: "satisfy property (i)" -> property (a)
 - Proof for Theorem 2: "it follows that n_sprunesn_t" -> formatting
 - Proof for Theorem 2 (case A): Lemma 1 only states that f(n_s) \leq f(n_t). I
 guess the equality comes from the fact that f(n_t) is the current minimum
 f-value?
 - Proof for Corollary 1: "which contains all algorithms if the family..." ->
 of the family
 - References: Corrêa et al 2018: a* algorithm -> A*

---

### Comment · Program_Chairs · 2020-09-14
**Final Decision: Accept**

Dear Authors,

Thank you very much for your submission. We are happy to inform you that we have decided to accept it and we look forward to your talk in the workshop. You will receive additional information per mail in the coming days.

Best,
The HSDIP'20 team

---

### Decision · Program_Chairs · 2020-09-30

Accept